# Identification of Differentially Expressed Genes in Resistant Tetraploid Wheat (*Triticum turgidum*) under *Sitobion avenae* (F.) Infestation

**DOI:** 10.3390/ijms23116012

**Published:** 2022-05-27

**Authors:** Xinlun Liu, Xudan Kou, Shichao Bai, Yufeng Luo, Zhenyu Wang, Lincai Xie, Pingchuan Deng, Hong Zhang, Changyou Wang, Yajuan Wang, Jixin Zhao, Wanquan Ji

**Affiliations:** 1State Key Laboratory of Crop Stress Biology for Arid Areas, College of Agronomy, Northwest A&F University, Xianyang 712100, China; 2013015441@nwsuaf.edu.cn (X.K.); baishichaobiology@163.com (S.B.); luoyufeng2020@nwafu.edu.cn (Y.L.); wzhenyu2018@163.com (Z.W.); 15100316205@163.com (L.X.); dengpingchuan@nwsuaf.edu.cn (P.D.); zhangh1129@nwafu.edu.cn (H.Z.); chywang2004@nwafu.edu.cn (C.W.); wangyj7604@nwafu.edu.cn (Y.W.); zhaojixin@nwafu.edu.cn (J.Z.); 2Shaanxi Research Station of Crop Gene Resources and Germplasm Enhancement, Ministry of Agriculture, Xianyang 712100, China

**Keywords:** aphid resistance, tetraploid wheat, *Sitobion avenae*, suppression subtractive hybridization, gene expression

## Abstract

The grain aphid *Sitobion avenae* (Fabricius) is one of the most destructive pests of wheat (*Triticum aestivum*). Deployment of resistant wheat germplasm appears as an excellent solution for this problem. Elite bread wheat cultivars only have limited resistance to this pest. The present study was carried out to investigate the potential of the tetraploid wheat (*Triticum turgidum*) variety Lanmai, which showed high resistance to *S. avenae* at both seedling and adult plant stages, as a source of resistance genes. Based on apterous adult aphids’ fecundity tests and choice bioassays, Lanmai has been shown to display antixenosis and antibiosis. Suppression subtractive hybridization (SSH) was employed to identify and isolate the putative candidate defense genes in Lanmai against *S. avenae* infestation. A total of 134 expressed sequence tags (ESTs) were identified and categorized based on their putative functions. RT-qPCR analysis of 30 selected genes confirmed their differential expression over time between the resistant wheat variety Lanmai and susceptible wheat variety Polan305 during *S. avenae* infestation. There were 11 genes related to the photosynthesis process, and only 3 genes showed higher expression in Lanmai than in Polan305 after *S. avenae* infestation. Gene expression analysis also revealed that Lanmai played a critical role in salicylic acid and jasmonic acid pathways after *S. avenae* infestation. This study provided further insights into the role of defense signaling networks in wheat resistance to *S. avenae* and indicates that the resistant tetraploid wheat variety Lanmai may provide a valuable resource for aphid tolerance improvement in wheat.

## 1. Introduction

Wheat, *Triticum aestivum* L., is an important food crop and one of the main sources of vegetable protein in human nutrition [1]. Aphids are the most important insect pests of world agriculture, damaging crops by removing phloem sap nutrients, vectoring numerous plant viruses and reducing photosynthetic efficiency [2]. The English grain aphid, *Sitobion avenae* (Fabricius), is one of the most destructive aphid species of wheat, seriously affecting the ecological and economic benefits in the wheat planting regions of China and some other grain-growing regions around the world [3,4,5,6,7]. Previous reports evidenced that *S. avenae* could cause significant changes in grain quality, infertility spikelet number, weight per grain and thousand-grain weight, and ultimately result in significant yield loss [4,8,9]. *S. avenae* infestation may cause severe crop losses in more than 63.25% of China’s wheat-growing areas and reduce wheat yield losses by as much as 33.45% [3,10]. The key characteristics of aphid, such as high mobility, short generation time and high reproductive rate, make it difficult to control. Consequently, large quantities of insecticides are used to control their colonization, which increases the food production costs and damages ecosystems in the short and long term, including non-target impacts on beneficial insects and high levels of insecticide resistance in aphid populations [11,12].

Cereal plants have evolved a wide range of defense mechanisms against aphid infestation (i.e., antibiosis, antixenosis and tolerance) [13,14,15]. So far, however, there are no elite hexaploid wheat cultivars with resistance to *S. avenae* [16], and only two *S. avenae* resistance genes have been identified. Therein, the *Ra**-**1*(*Sa1*) is from *Triticum*
*durum* L. (C273) [17], and *Sa2* is from the synthetic wheat line (98-10-35) [18]. Most commercial bread wheat cultivars have a very low resistance to aphid pests [16,19,20,21]. However, tetraploid wheat varieties (*Triticum turgidum* L.) have been shown to be highly resistant to wheat aphids, including English grain aphid (*S. avenae*), greenbug (*Schizaphis graminum* Rondani) and Russian wheat aphid (*Diuraphis noxia* Morduilko) [19,22,23,24,25], providing a useful source of material to analysis of resistance mechanisms. Aphid resistance genes derived from these tetraploid wheat accessions will be useful to broaden the basis of aphid resistance in hexaploid wheat breeding [23].

Although few commercial wheat cultivars are available against aphids, some defense genes are specifically induced post aphid infestation in wheat. Previous reports evidenced that *S. avenae* feeding induced the expression of several genes involved in jasmonic acid (JA), salicylic acid (SA) and ethylene (ET), signaling defense pathways in wheat [26,27,28]. The transcript levels and enzyme activities of reactive oxygen species (ROS) scavengers and oxidative stress response proteins were induced after the infestation of *S. avenae* [27,28]. Similarly, antioxidative enzymes, such as superoxide dismutase, glutathione reductase and ascorbate peroxidase, were significantly enhanced after *D. noxia* infestation in the resistant wheat (*T. aestivum*) cultivar [29]. In resistant diploid wheat, oxidative stress response proteins and NBS–LRR-like proteins were significantly upregulated in response to *S. avenae* feeding [30]. Proteomic analysis revealed that the majority of proteins upregulated under *S. avenae* infestation involved in metabolic processes, photosynthesis and ATP synthesis in both aphid-resistant bread wheat and aphid-resistant diploid wheat [26,30]. Similarly, Botha et al. [31] elucidated that most of the differentially regulated genes after *D. noxia* infestation, in near-isogenic wheat lines with different *Dn* genes, belonged to the carbohydrate metabolism and energy metabolism and well-represented genes involved in starch synthesis, photosynthesis or chloroplast-related proteins. Additionally, Reddy et al. [32] uncovered that the differentially expressed transcripts in *S. graminum*–wheat interactions were grouped into six major functional categories: ‘Misc’, mediating amino acid metabolism, transport, protein-related, stress and signaling groups. Moreover, membrane-bound receptor kinases, calcium signaling and other early biotic stress signaling pathways were activated by *S. graminum* in resistance genotypes with the *Gb3* gene [32].

However, few studies have attempted to identify the defense responses of tetraploid wheat to cereal aphid infestation. Our previous study revealed that the tetraploid wheat variety Lanmai (Shaanxi zhashui) showed high resistance to *S. avenae* at both seedling and adult plant stages [19]. In this study, we integrated gene expression profiling by suppression subtractive hybridization (SSH) and RT-qPCR to reveal the responses of tetraploid wheat leaves to *S. avenae* infestation. In addition, we evaluated the preference and the fecundity of *S. avenae* feeding on the resistant variety Lanmai and the susceptible variety Polan305 seedlings and investigated the reduction ratio of thousand-grain weight caused by *S. avenae* infestation. This study will expand the understanding of the molecular mechanisms underlying the defense response of tetraploid wheat to cereal aphids.

## 2. Results

### 2.1. SSH Library Construction

A total of 600 positive clones were identified from the SSH-cDNA library constructed from aphid-infested and non-infested seedlings, of which 586 expressed sequence tags (ESTs) met the quality requirements after sequence cleaning. The high-quality ESTs were assembled into 134 contigs using the BioEdit Sequence Alignment editor. The length of contigs ranged from 146 bp to 1210 bp, with an average length of 399 bp, and most of these contigs were between 209 bp and 497 bp. All these 134 unique ESTs were submitted to the EST database of GenBank (http://www.ncbi.nlm.nih.gov/dbEST) (GenBank ID: JK716033–JK716166) (accessed on 11 December 2011) (Appendix A). This library was used to elucidate transcript expression changes and identify transcripts, which show differential responses triggered by *S. avenae* infestation on the tetraploid wheat variety Lanmai.

### 2.2. Annotation and Gene Ontology Analysis

BLASTx/BLASTn analysis of all the 134 aphid-induced ESTs of the SSH library against the non-redundant protein sequences (nr) database and URGI wheat genome database was used to identify the homology of these unigenes with the threshold E-value < 1 × 10^−5^ (Appendix A). BLAST results showed that 112 ESTs (83.58%) were homologous to previously characterized proteins of *Triticum* crops, and 9 ESTs (6.71%) were homologous to genes with hypothetical proteins or unnamed proteins in the protein databases. However, all the 134 ESTs were homologous to previously characterized proteins in the URGI wheat genome database except for JK716147. They have been identified to have functional annotations related to various biological processes, involved in defense and defense signaling (14.9%), photosynthesis and energy (14.9%), protein synthesis (14.2%), protein degradation (6.7%), metabolism (11.9%), signal transduction (9.7%), transport (9.0%), transcription (6.0%), cell organization and division (3.0%) and cell wall (2.2%) (Figure 1 and Appendix A)

Gene ontology (GO) is extensively used to describe gene characteristics and functions of gene products. To further characterize the function of ESTs, a GO enrichment analysis was conducted (Figure 2 and Appendix A), and the results showed that, among the 134 aphid-induced ESTs, 106 ESTs (79.10%) were assigned to GO ontology in Blast2GO and classified into three main ontology categories: biological processes, molecular function and cellular components. Among the cellular component category, the ESTs induced by *S. avenae* were mainly enriched in ‘cell’ (20), ‘cell part’ (20), ‘macromolecular complex’ (15) and ’organelle’ (8). Under the molecular function category, the largest proportion of ESTs activated by *S. avenae* were re-enriched in ‘binding’ term (62), the majority of ESTs induced by *S. avenae* were re-enriched in ‘catalytic activity’ term (44) and the rest of ESTs induced by *S. avenae* were re-enriched in ‘transporter activity’ (6) and ‘structural molecule activity’ (6). Within the biological processes category, the ESTs activated by *S. avenae* were mainly enriched in ‘metabolic process’ (58), ‘cellular process’ (44), ‘establishment of localization’ (10) and ‘localization’ (10). In addition, GO enrichment analysis showed that ESTs activated by *S. avenae* were enriched in the metabolic process (58), cellular metabolic process (38) and transferase activity (9). The enzymes activated by *S. avenae* infestation were mainly involved in catalytic activity, carbohydrate metabolic process, protein metabolic process, L-arabinose metabolic process, structural constituent of ribosome, oxidoreductase activity, protein kinase activity and defense response. These results showed that the differentially expressed ESTs under aphid-infested conditions may regulate protein-coding genes involved in some important biological processes.

### 2.3. Gene Expression Analysis

To validate the results of SSH in wheat and further understand the aphid–wheat interaction mechanism, we accessed the expression of 30 representative ESTs by RT-qPCR in the resistant variety Lanmai and susceptible variety Polan305 under aphid infestation, using beta-*actin* as the reference gene (Figure 3 and Appendix A). The 30 aphid-induced ESTs were selected from different functional groups, and the summary of the selected ESTs sequence comparisons and functional annotation is presented in Appendix A. RNA samples were collected from leaves of Lanmai and Polan305 at 0, 12, 24, 48, 72 and 96 h post-aphid infestation (hpi).

*S. avenae* infestation has differential effects on many genes associated with light harvesting and photosystems. For example, two putative genes involved in photosystem II (i.e., psbP-like protein 1 (Appendix A) and light-harvesting complex-like protein OHP2 (Appendix A)), two genes with roles in the Calvin cycle (i.e., ribulose-1,5-bisphosphate carboxylase/oxygenase large subunit (Appendix A) and phospholipase A1-II 7-like (Appendix A)) and one gene encoding the protein CHUP1 (Appendix A) were higher expressed in Polan305 seedlings than that in resistant variety Lanmai throughout the experiment. However, a few genes involved in photosynthesis were significantly regulated in the *S. avenae*-infested plants. A wheat gene encoding a chlorophyll a-b binding protein (Figure 3A) was up-regulated at 24 hpi, and the highest expression was observed at 96 hpi with 266.45- and 98.44-fold upregulation in Lanmai and Polan305, respectively. The transcript levels of a wheat gene, encoding protein thylakoid formation 1 (Appendix A), continued to slightly increase from 0 hpi to 96 hpi after *S. avenae* infestation in Lanmai, whereas a significant increase in gene expression was detected in Polan305 at 24 hpi. The transcript levels of fructose-1,6-bisphosphatase (Appendix A), which plays a role in the Calvin cycle, were slightly upregulated at 48 and 72 hpi only in the resistant wheat variety Lanmai.

The transcriptional profiles of some genes involved in L-arabinose and glucose metabolism were also investigated. *S. avenae* infestation consistently induced higher transcript levels of the gene encoding alpha-L-arabinofuranosidase 1 (Appendix A) in susceptible wheat variety Polan 305 compared to in resistant Lanmai, except at 48 hpi. Meanwhile, the transcript level of this gene was the highest at 48 hpi in Lanmai and was upregulated by 8.94-fold compared with the non-infested control. The expression levels of the malate dehydrogenase gene (Figure 3B) with a role in the TCA process were also significantly upregulated in Polan305 at 24 hpi after *S.*
*avenae* infestation; however, its expression was observed to be almost uniform in Lanmai seedlings in all the time series. The expression levels of a putative gene, glyceraldehyde-3-phosphate dehydrogenase 1 (Appendix A), involved in glycolysis were higher in seedlings of Polan305 than that in Lanmai in the whole experiment.

*S. avenae* infestation positively affected the protein synthesis process as well as many genes associated with protein synthesis and post-translational modification-associated genes in wheat, such as probable protein phosphatase 2C (Appendix A), which may be involved in protein post-translational modification, was significantly up-regulated in Lanmai after *S. avenae* infestation; however, a slight downward trend was observed in Polan305 compared to the non-infested control. Moreover, the expression levels of 60S ribosomal protein (Figure 3C), protein translation factor SUI1-like protein (Figure 3D) and translation machinery-associated protein 22-like (Appendix A) with roles in the protein synthesis were also significantly upregulated after *S. avenae* infestation. The highest increments in expression levels of 60S ribosomal protein (Figure 3C) and protein translation factor SUI1-like protein (Figure 3D) in Lanamai seedlings at 48 hpi (3.95-fold) and 24 hpi (2.95-fold), respectively, compared to the non-infested control. Besides, the transcript abundance of the two genes was not significantly increased in Polan305 seedlings in comparison with the non-infested control. Furthermore, *S. avenae* infestation induced higher transcript levels of translation machinery-associated protein 22-like (Appendix A) and chlorophyllide, an oxygenase (Appendix A) involved in protein targeting only at 48 hpi compared to the non-infested control, with 2.49- and 2.10-fold upregulation in the resistant wheat variety Lanmai and susceptible wheat variety Polan 305, respectively.

The genes associated with transport progress were up- or downregulated in response to *S. avenae* infestation. After *S. avenae* infestation, the expression level of the probable aquaporin TIP2-2 (Appendix A) was significantly upregulated by 7.37-fold at 12 hpi and then decreased sharply in Polan305 compared to non-infested control, whereas the expression of probable aquaporin TIP2-2 decreased continuously from 0 hpi to 96 hpi in Lanmai. However, compared to the non-infested control, fluctuating increases in transcriptional accumulation of two wheat genes, namely probable anion transporter 5 (Appendix A) and probable calcium-transporting ATPase 9 (Appendix A), were observed in Lanmai and Polan305; the highest expression levels of probable calcium-transporting ATPase 9 occurred at 96 hpi in Lanmai (upregulated 4.12) and Polan 305 (upregulated 2.45). The transcript levels of MADS-box transcription factor (Figure 3E) and NAC domain-containing protein 41-like (Figure 3F) with roles in transcription were also much more strongly upregulated in Lanmai than in Polan305 after *S. avenae* infestation. The highest increments in MADS-box transcription factor expression levels occurred at 24 hpi (19.15-fold) and 48 hpi (3.02-fold) in Lanamai and Polan305 seedlings, respectively, compared to the non-infested control. The highest increments in the expression level of NAC domain-containing protein 41-like occurred at 48 hpi (3.15- and 2.85-fold in Lanmai and Polan305, respectively) compared to the control. *S.*
*avenae* infestation induced the accumulation of the expressed transcript of a wheat gene encoding serine/threonine protein phosphatase 2A (Appendix A), which gradually increased to a maximum and then decreased in Polan305 and Lanmai, and the largest gene upregulation in Polan305 and Lanmai was 1.90- and 2.23-fold, respectively. The expression pattern of a wheat gene encoding RAN-binging protein 1 (Appendix A) was consistent with that of serine/threonine protein phosphatase 2A (Appendix A) in Lanmai; however, transcript accumulation from 12 hpi to 96 hpi was observed in Polan305, and the maximal upregulation of this gene was 1.95- and 2.30-fold in Polan305 and Lanmai, respectively.

*S. avenae* infestation significantly upregulated the expression of genes known to be involved in SA and JA pathways and the antioxidative system, and the transcript levels of some defense genes were much higher in resistant wheat variety Lanmai than that in susceptible wheat variety Polan305. For example, for the putative lipoxygenase 2.1 gene (Figure 3G) involved in JA biosynthesis, the highest expression levels occurred at 24 hpi (7.42-fold) and 72 hpi (1.82-fold) in Lanamai and Polan305 seedlings, respectively, compared to the non-infested control. The highest expression levels of the putative protease inhibitor Bsi1 gene (Figure 3H), a JA-responsive defense gene, occurred at 96 hpi in both Lanamai (4.39-fold) and Polan305 (1.72-fold). The highest expression levels of the wPR4a gene (Figure 3I) and a wheat gene encoding putative BPI/LBP family protein (Figure 3J), responding to SA, were observed at 48 hpi (58.84-fold) and 24 hpi (16.90-fold) in Lanmai seedlings infested by *S. avenae* compared to the non-infested control. After *S. avenae* infestation, the highest expression levels of monodehydroascorbate reductase (MDAR) (Figure 3K) with roles in the ascorbate-glutathione cycle and S-adenosylmethionine decarboxylase (Appendix A) involved in polyamine metabolism were upregulated by 4.39-fold and 2.30-fold at 24 hpi in Lanamai compared to the non-infested control, respectively. Nevertheless, the wheat gene encoding multiprotein-bridging factor 1a-like (Figure 3L), involved in ethylene metabolism, was significantly downregulated in Lanmai and Polan305 by *S. avenae* infestation.

### 2.4. Evaluation of S. avenae Aphid Damage, Adult Fecundity and Preference

Wheat plants were seriously infested by numerous aphids, the thousand-grain weight of both Lanmai and Polan305 was significantly lower than those of the control and the reduction ratio of thousand-grain weight of wheat was 42.16% and 30.71%, respectively (*p* < 0.0001; Figure 4A).

To elucidate whether the mechanism resistance of Lanmai is antibiosis, adult fecundity was performed. The results showed that, during the first 10 days after *S. avenae* infestation, the number of nymphs feeding on Lanmai seedlings and Polan305 seedlings was not significantly different (*p* ≥ 0.0670; Figure 4B). However, from the 14th day, both adults and nymphs feeding on Lanmai seedlings were significantly lower than Polan305 seedlings (*p* < 0.05; Figure 4B). A disparity increase in the number of *S. avenae* feeding on Lanmai seedlings and Polan305 seedlings was observed over time.

To determine whether *S. avenae* showed the infestation preference for either the resistant wheat variety Lanmai or susceptible wheat variety Polan305, 30 apterous adult aphids were presented with a choice between resistant and susceptible wheat seedlings. The results of the choice assay showed that the number of aphids choosing Lanmai seedlings and Polan305 seedlings was not significantly different in the first 9 h (*p* ≥ 0.5389; Figure 4C). However, from the 12th hour, the number of aphids choosing Polan305 seedlings was significantly higher than that of Lanmai seedlings (*p* < 0.05; Figure 4C), and from this point, there was a significantly higher number of aphids on Polan305 seedlings than Lanmai seedlings over the time course.

## 3. Discussion

Previous reports evidenced that cereal plants’ main mechanisms were responsible for imparting resistance to aphids including antibiosis, antixenosis and tolerance [13,14,15]. Antibiosis is the adverse impact of resistant plants on the survival, growth, longevity or fecundity of insects [13,14,15]. Antixenosis is the non-preference reaction of insects to resistant plants, leading to delay, the acceptance of plants as hosts or prevention of insect infestation [13,15]. Tolerance enables resistant plants to maintain growth and productivity, despite harboring insect numbers similar to those observed on susceptible plants [15]. These authors revealed that different resistant wheat varieties (lines) have significant effects on nymph mortality adult longevity and adult fecundity, and the fecundity of *S. avenae* has been used to evaluate the adaptability of aphids to the host crops [33,34,35]. Thousand-grain weight is significantly affected by S. *avenae* [8,9], and this trait was used to investigate wheat tolerance to aphids in previous reports [33,36]. H Havlíčková (1997) [37] also proved that the losses in seed weight caused by the cereal aphids (*M. dirhodum*, *R. padi* and *S. avenae*) were positively correlated with aphid densities. In the present study, the resistant wheat variety Lanmai displayed both antixenosis and antibiosis modes of resistance. In preference assays, *S. avenae* adults exhibited a strong preference for the susceptible variety Polan305 instead of Lanmai seedlings (Figure 4C). Host selection by adult aphids is usually the first stage of colonization and plays an important role in the determination of aphid populations in the field. *S. avenae* adults avoid Lanmai, which is likely a major part of their aphid resistance. In the antibiosis test, we found that the two wheat varieties had different impacts on the propagation of *S. avenae*. Compared with the aphids infested on Polan305, the number of aphids infested on Lanmai was significantly lower, suggesting that *S. avenae* infesting on Lanmai put in a worse performance than those on Polan305, and thus, Lanmai was more highly resistant to aphids than Polan305. However, the thousand-grain weight of both Lanmai and Polan305 plants suffered a significant loss after *S. avenae* infestation, which indicated that a low level of tolerance to *S. avenae* was found in both Lanmai and Polan305.

Photosynthesis produces saccharides, a key source of energy and structural elements for cells. Repairing or performing de novo synthesis of photosystem proteins is a main mechanism of the plant tolerance to pests [2]. In general, when plants are attacked by herbivores, photosynthesis-related genes may be upregulated to compensate for the loss of nutrition caused by the aphid attack [2], and tolerant plants exhibit upregulated photosynthetic capacity [31]. It has been also reported that differential expression of metabolic processes or photosynthetic genes was also observed, such as *Myzus persicae* infesting on celery [38], *M. nicotianae* infesting on tobacco [39], *M. sacchari* infesting on sorghum [40], *S. avenae* infesting on diploid wheat [30] and *D. noxia* infesting on bread wheat [31,41], where each promoted a significant upregulation of such genes; additionally, some photosynthesis genes were downregulated in bread wheat after infestation by *S. avenae* and *S. graminum* [26,27].

In the present study, several genes directly associated with the Calvin cycle, photosystem II and metabolism were differentially regulated in both genotypes following *S. avenae* infestation. For example, the photosystem II light-harvesting gene, chlorophyll a-b binding protein 1C (Figure 3A), was strongly upregulated in *S. avenae*-infested Lanmai and Polan305 plants. Similarly, protein thylakoid formation 1 (Appendix A) and phospholipase A1-II 7-like (Appendix A) were upregulated in response to *S. avenae* infestation in both Lanmai and Polan305. However, only seen in the resistant wheat variety Lanmai, *S. avenae* feeding slightly upregulated the abundance of fructose-1,6-bisphosphatase (Appendix A) with roles in the Calvin cycle. Conversely, the expression levels of complex-like protein OHP2 (Appendix A), ribulose-1,5-bisphosphate carboxylase/oxygenase large subunit (Appendix A), malate dehydrogenase (Figure 3B) and psbP-like protein 1 (Appendix A) were only slightly increased in the susceptible variety Polan305 plants after *S. avenae* infestation. Similarly, Guan et al. [30] revealed that ribulose-1, the 5-bisphosphate carboxylase/oxygenase large subunit, was upregulated in response to *S. avenae* infestation in wild Einkorn wheat. Moreover, Park et al. [42] claimed that the chlorophyll a-b binding protein was upregulated in a susceptible sorghum line and downregulated in a resistance line by greenbug feeding. Furthermore, Botha et al. [31] evidenced that the chlorophyll a-b binding protein was only significantly upregulated in the antibiotic Tugela-*Dn1*, while well-represented genes involved in carbon flux and photosynthesis were upregulated only in the tolerant Tugela-*Dn2*. In the current study, almost all genes involved in the Calvin cycle, photosystem II and metabolism were upregulated in the susceptible variety Polan305, while half of them were observed to be downregulated in Lanmai after being infested by the *S. avenae*. The result is consistent with the loss of thousand-grain weight caused post *S. avenae* infestation and demonstrates that the resistant wheat variety Lanmai does not exhibit aphid tolerance.

Changes in protein factors are an important part of plant defense mechanisms [26]. Zhang et al. [27] recorded that the DEGs induced by *S. avenae* were mainly enriched in biological process classes, such as the protein modification process, protein phosphorylation and phosphorus metabolic process. In our study, within the biological process category, ESTs induced by *S. avenae* were mainly enriched in the photosynthesis light reaction, Calvin cycle, protein synthesis and protein post-translational modification processes. Ten and five genes were related to protein synthesis and protein post-translational modification processes, respectively, which were in response to *S. avenae* infestation. Among them, one targeting chloroplast protein gene (chlorophyllide, an oxygenase, chloroplastic-like protein (Appendix A)), three genes associated with protein synthesis (translation machinery-associated protein 22-like Appendix A), 60S ribosomal protein L10-1 (Figure 3C) and protein translation factor SUI1-like (Figure 3D)) and one protein post-translational modification gene (probable protein phosphatase 2 (Appendix A)) were more upregulated in the resistant wheat variety Lanmai than in the susceptible variety Polan305. Similar results have been reported, with genes regulating transcription and protein synthesis being upregulated by aphid feeding [26,43,44].

Plants infested by aphids would induce a series of defense responses, including transcriptional regulation, plant hormone signal transduction and the expression of defensive genes [2]. Transcription factors are important regulators of plant defense response and have been reported to be involved in plant–aphid interactions. Transcription factor genes *MYBs* were highly activated in wheat and Chrysanthemum in response to aphid infestation [45,46]. In this study, MADS-box transcription factor (Figure 3E) and transcriptional activator NAC domain-containing protein (Figure 3F) were observed to be highly upregulated in the resistant wheat variety Lanmai. MADS-box TFs were observed in antibiotic and antixenotic genotypes in response to *D. noxia* infestation [31], and the NAC1 stress response protein was strongly upregulated in wheat plants infested with *R*. *padi* and *S*. *graminum,* whereas there were no significant differences between *S*. *avenae*-infested and healthy plants [44].

Hormone signaling plays a critical role in plant immunity. Jasmonic acid (JA), salicylic acid (SA) and ethylene (ET) are the major phytohormones involved in aphid-induced defense responses [47]. Several studies have characterized that aphid infestation strongly induces the expression of genes involved in SA-dependent pathway in wheat, barley, sorghum and *Arabidopsis thaliana*, and increased the expression of pathogenesis-related (PR) genes (i.e., PR1, chitinases and β-1,3-glucanase) associated with the signaling pathway [27,48]. In the current study, the SA-responsive wPR4a gene (Figure 3I) and putative BPI/LBP family protein gene (Figure 3J) [49], as well as the JA-responsive Lipoxygenase 2.1 gene (Figure 3G) and the putative protease inhibitor Bsi1 gene (Figure 3H) were significantly upregulated by *S*. *avenae* infestation. Similarly, in wheat, the expression of *LOX1* was significantly induced by *S*. *avenae* and *S*. *gramin* [44]. In barley, *S*. *avenae* infestation significantly induced the expression of lipoxygenase 2 [50]. Overexpression or suppression of *LOX2.2* did not affect aphid settling or plant lifespan, but lines overexpressing *LOX2.2* supported lower aphid numbers and showed upregulation of some other jasmonic acid (JA)regulated genes compare to antisense plants [51]. Recently, Zhang et al. (2019) [27] found that five LOX (2.8- to 8.8-fold) genes were significantly upregulated by both *S*. *graminum* and *S*. *avenae* infestation in winter wheat. Pingault et al. (2021) [52] also found that 7 out of 13 *LOX*-annotated genes encoding 9-lipoxygenases were significantly induced in maize by corn leaf aphid feeding. In previous studies, two types of barley proteinase inhibitors (chymotrypsin and trypsin inhibitors) induced by *S. graminum* and *R. padi* decreased the survival of *R. padi* [53]. A putative proteinase inhibitor and protease inhibitor CI2c were upregulated in resistant barley lines but not in susceptible lines, affecting aphids negatively [48,54]. Zhang et al. [27] demonstrated that the expression levels of three PI genes in wheat were strongly induced by *S. graminum* and *S. avenae* infestation.

EIN2 is required for directed translational regulation of ethylene signaling in Arabidopsis [55]. Recently, Pingault et al. (2021) [52] unveiled that two members of the EIN gene family showed increased expression in the susceptible maize genotype compared to the resistance maize genotype at both 0 and 24 hpi in leaves and roots. In the current study, we demonstrated that multiprotein-bridging factor 1a-like (Figure 3L) involved in ethylene biosynthesis was significantly reduced in Lanmai and Polan305 by *S. avenae* infestation. Ethylene response factor (ERF) is a large class of AP2/EFR domain-containing TF. Additionally, 3 of the 13 annotated ERFAP2 transcription factors were exclusively upregulated at 24 hpi in the resistant maize genotype, while 9 were upregulated in both resistant and susceptible maize genotype and tissues [52]. Similarly, *S. graminum* infestation significantly upregulated the ET signaling pathway of wheat, as 2 ACS and 3 ACO genes were involved in the ET signaling pathway, and 11 genes encoding ethylene-responsive transcription factors involved in ET biosynthesis were significantly upregulated, but *S. avenae* feeding only upregulated 1 ACO gene and 2 genes encoding ethylene-responsive transcription factors [27].

Ascorbic and glutathione are the most crucial components of the antioxidative system in the leaves of the higher plants [56]. Monodehydroascorbate reductase (MDAR) is one of the key enzymes operating within the ascorbate-glutathione cycle [57]. In this study, MDAR (Figure 3K) was found to be significantly upregulated in the resistant wheat variety Lanmai compared to the susceptible wheat variety Polan305 at 24 and 48 hpi. Similarly, Sytykiewicz [58] ascertained that the aphid-infested resistant maize varieties plants reached higher expression levels in all four types of monodehydroascorbate reductase genes (*MDHAR1*, *MDHAR2*, *MDHAR3* and *MDHAR4*) compared to susceptible maize varieties seedlings, with *MDHAR1* remarkably accumulated in tissues of less resistant maize varieties. However, Reddyet et al. [32] elucidated that greenbug feeding had caused a downregulation of monodehydroascorbate reductase (MDAR) in the resistant bulk compared to susceptible bulk at 24 and 48 hpi in near-isogenic wheat lines. Hence, this hinted that MDARs may played a complex role in plant responding to pest with influence of genetic background.

## 4. Materials and Methods

### 4.1. Plant Materials and Growth Conditions

Two tetraploid wheat varieties (i.e., Lanmai (Shaanxi, Zhashui, China) and Polan305, *Triticum turgidum)* were selected for study, representing *S. avenae*-resistant and -susceptible wheat genotypes, respectively. Lanmai was used for SSH cDNA library construction, and Lanmai and Polan305 were used for validation of shortlisted genes in gene expression. The seeds were obtained from the College of Agronomy, Northwest A & F University, China. Wheat seedlings were grown in plastic pots (12.5 cm in diameter) with planting soil (containing peat and vermiculite at a 3:1 ratio). The plants were grown in a growth chamber at 22 ± 1 °C with 50–60% relative humidity and a photoperiod of 16/8 h (day/night), with a photosynthetic photon flux density of 150 μmol photons m^−2^ s^−1^.

### 4.2. Aphids

A single viviparous apterous female adult *S. avenae* was collected from an infested wheat field in Yangling, Shaanxi Province, China. The aphid was reared on the highly susceptible wheat cultivar plants (Xiaoyan 6) in a growth chamber with conditions as described above.

### 4.3. Aphid Infestation Biotests

Both Lanmai and Polan305 plants for the experiments were used at the two-leaf developmental stage (7-day-old plants), and each plant was infested with 10 apterous *S. avenae* adults for 12, 24, 48, 72 and 96 h. *S. avenae* non-infested samples were collected at 0 hpi as control plants. All plants were covered with the transparent plastic cylinder (6 cm in diameter, 20 cm long) capped with one layer of the nylon mesh at the start of the experiment, and the addition of *S. avenae* was staggered so that all plant samples for SSH cDNA library construction and gene expression were harvested at the same time. After the termination of each series of the aphid infestation, all the aphids were removed and then the leaf samples from the aphid feeding sites of each plant were harvested. All the samples were immediately frozen in liquid nitrogen and stored at −80 °C until further processing for RNA extraction. Nine plants were infested in each treatment, and three samples were collected from each replicate to form three replicates.

### 4.4. RNA Isolation, Suppression Subtractive Hybridization and Library Construction

Total RNA of the resistant wheat variety Lanmai was extracted from aphid-infested and non-infested (control) wheat seedlings using TRIzol reagent (Invitrogen, Carlsbad, CA, USA), following the manufacturer’s protocol. The quantity and quality of RNA were evaluated by measuring absorbance at 260 nm and 1% agarose gel electrophoresis. A forward subtractive cDNA library was constructed from Lanmai seedlings infested with *S. avenae* by the suppression subtractive hybridization (SSH) approach, using Clontech PCR Select™ cDNA subtraction kit (Clontech, Mountain View, CA, USA), according to manufacturer’s protocol. For the construction of the library, 2 µg of a pool of mRNAs (equal amounts of each mRNA were isolated from Lanmai plants infested by *S. avenae* at 24, 48 and 72 h after infestation) was used as the ‘tester’; also for the ‘Driver’ sample, 2 µg of total mRNAs were isolated from Lanmai with no *S. avenae* infestation (0 h). Differentially expressed cDNAs with different adaptors (1A and 2R) at two ends were selectively amplified by PCR. A secondary PCR was performed with nested primers to enrich differentially expressed sequences and further reduce background PCR products. The secondary PCR products of the subtracted library were ligated to the pGEM-T easy vector (Promega, Madison, WI, USA) and subsequently transformed to Escherichia coli strain *DH5a*. The transformed cells were placed onto *X-gal*, *IPTG* and ampicillin-supplemented LB medium at 37 °C overnight. Only white colonies were randomly selected and checked with T7/SP6 primers for the size of cDNA inserts.

### 4.5. Sequencing and Sequence Analysis

A total of 600 clones, possessing an insert size of 200 bp and above, were sequenced by Sangon Bioengineering Inc. (Shanghai, China) using the Sanger dideoxy method. In all sequenced ESTs, vector and adaptor sequences were removed using VecScreen software (http://www.ncbi.nlm.nih.gov/tools/vecscreen/ accessed on 11 December 2011). ESTs with low quality and short sequences (<100 bp) were excluded. The CAP3 program (http://www.doua.prabi.fr/software/cap3 accessed on 11 December 2011) was used for clustering the ESTs into contigs and singlets. All ESTs were annotated by Blastx or Blastn with both the NCBI database (http://blast.ncbi.nlm.nih.gov/Blast accessed on 11 December 2011) and the URGI wheat genome database (https://urgi.versailles.inra.fr accessed on 11 December 2011). Concurrently, the gene ontology annotation was performed with the Blast2GO (https://www.blast2go.com/blast2go-pro accessed on 11 December 2011) software tool according to plant-specific gene ontology terms [59]. Furthermore, the functional annotation was performed by MapMan, which allows for attributing DEGs to functional pathways [60]. All EST sequences isolated in this study were submitted to the GenBank database and are shown in Appendix A.

### 4.6. Expression Profiling of Selected ESTs Using Quantitative Real-Time PCR (qRT-PCR)

Quantification of the target genes mRNA abundance in leaves of wheat seedlings was performed with the application of the real-time qRT-PCR technique, using Applied Biosystems Q7 Real-Time PCR system (Thermo Fisher Scientific) with SYBR Premix Ex Taq (Takara). *β-actin* was used as the internal reference gene. Total RNA was extracted from Lanmai and Polan305 seedlings at 0, 12, 24, 48, 72 and 96 h after *S. avenae* infestation using RNAiso Plus (Takara). First-strand cDNA was synthesized using the PrimeScript™ RT reagent Kit (TaKaRa) following the manufacturer’s instructions. Each reaction was performed in triplicates, and the relative gene expression was calculated with the 2^−∆∆CT^ method [61]. The primers used in the study are listed in Appendix A.

### 4.7. Evaluation of S. avenae Damage, Performance and Preference

Evaluation of the tolerance of wheat varieties (Lanmai and Polan305) to *S. avenae* infestation was carried out under field conditions from 2019 to 2021 in Yangling, Shanxi, China. Wheat plants were planted in hill drops of 1 m in length and 24 cm between the rows. Twenty seeds were planted in two rows and two clusters of wheat plants were planted in each row, with five seeds per hole, and arranged in plum blossoms. Fifty apterous aphids were placed on each plant at the booting stage. Three replicates were performed in a randomized design in the preventing insect nets. Lanmai and Polan305 non-infested plants were used as controls and planted in another preventing insect nets. Thousand-grain weight was measured from the harvested seeds of each replicate. The damage degree of *S. avenae* was evaluated by the weight reduction ratio of 1000 grains of wheat.

To determine the fecundity of *S. avenae* infesting on Lanmai and Polan305 seedlings (7-day-old plants), each plant was infested with one apterous *S. avenae* adult and covered with a transparent plastic cylinder (6 cm in diameter, 20 cm long). The numbers of both adult aphids and nymphs per plant were counted at 3, 7, 10, 14 and 21 days post infestation (nine replicates over three experiments).

To examine *S. avenae* preference, Lanmai and Polan305 seedlings were grown in pots (12.5 cm in diameter by 9 cm in depth). Seeds of Lanmai and Polan305 were interplanted near the perimeter of the pot and arranged equidistant from other seeds and the center of each pot (approximately 4.0 cm between plants and 4.5 cm from the center) with nine replicates. Thirty apterous adult aphids were transferred to a plastic disc in the center of each pot, which was approximately 1.0 cm away from each seedling (7-day-old plants). The number of adult aphids per wheat seedling was counted at 3, 6, 9, 12, 24, 48, 72, 96 and 120 h post aphid introduction (nine replicates over three experiments).

### 4.8. Statistics

All data are presented as the mean of at least three independent replicates. RT-qPCR results, weight loss per thousand grains and preference and fecundity of *S. avenae* were analyzed using the GraphPad Prism software. The two-way ANOVA followed by Sidak’s multiple comparisons test was used to compare the means of all our data.

## 5. Conclusions

In summary, our results indicate that the underlying resistance mechanisms of the tetraploid wheat variety Lanmai after *S. avenae* infestation mainly include both antixenosis and antibiosis. Expression pattern analysis demonstrated that the resistant wheat variety did not display the same expression levels of aphid-responsive genes as the susceptible wheat variety Polan305 during *S. avenae* infestation. The results suggest that oxidative stress response, defense response and protein synthesis play important roles in conferring resistance to *S. avenae* in Lanmai. This study indicates that Lanmai is valuable source of aphid resistance, which can be introgressed into a wide range of resistant lines for the production of bread wheat with durable resistance to *S. avenae.*

## Figures and Tables

**Figure 1 ijms-23-06012-f001:**
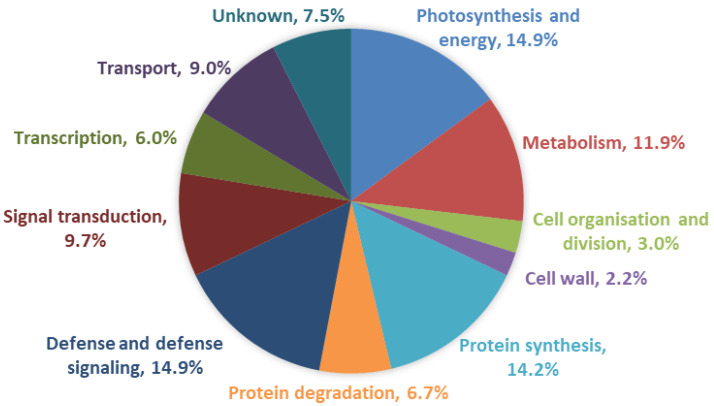
Functional classifications of SSH-derived ESTs homologous to known functional proteins.

**Figure 2 ijms-23-06012-f002:**
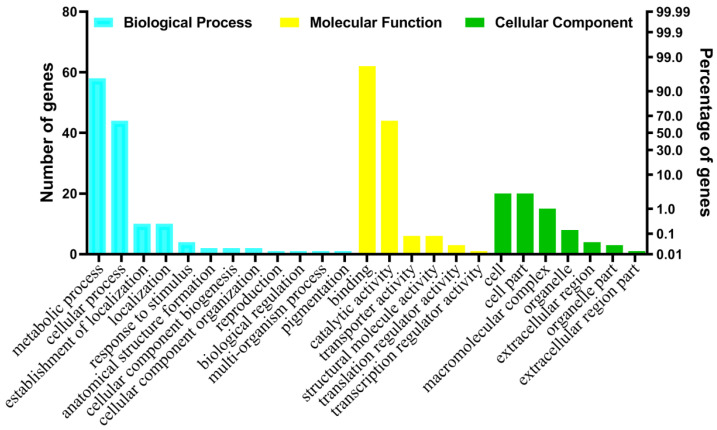
Functional classification of *S. avenae*-induced ESTs in tetraploid wheat seedlings identified from suppression subtractive hybridization library (SSH). Classification of ESTs based on biological process, cellular components and molecular function using Blast2GO software (http://blast2go.org) (accessed on 26 August 2021).

**Figure 3 ijms-23-06012-f003:**
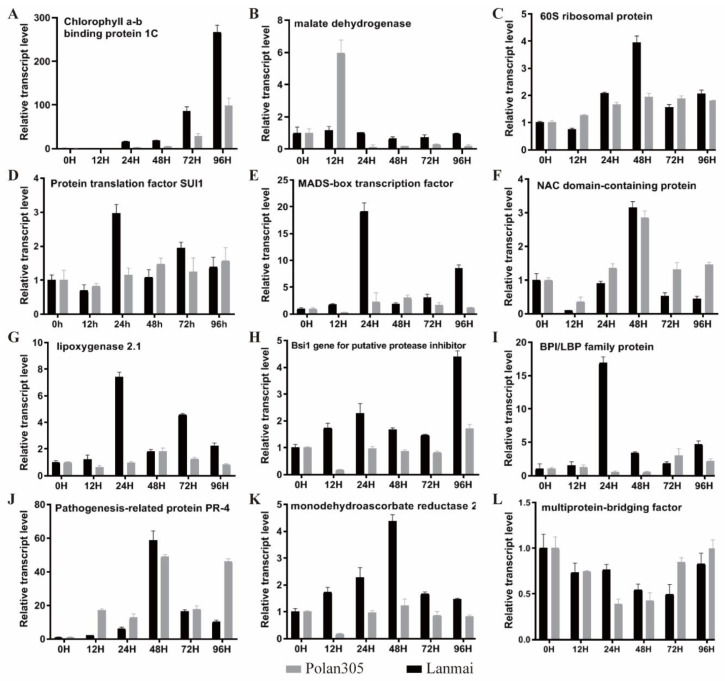
Expression profiles of 12 representative ESTs in seedlings of the resistant (Lanmai) and susceptible (Polan305) varieties under S. avenae infestation at different time points. RT-qPCR was used to detect the expression levels of the candidate genes. TaActin was used as an internal control. Each value represents the mean ± SE of three biological replicates. chlorophyll a-b binding protein (**A**), malate dehydrogenase (**B**), 60S ribosomal protein (**C**), SUI1-like protein (**D**), MADS-box tran-scription factor (**E**), NAC domain-containing protein (**F**), lipoxygenase 2.1 (**G**), Bsi1 gene for puta-tive protease inhibitor (**H**), BPI/LBP family protein (**I**), Pathogenesis-related protein PR-4 (**J**), monodehydroascorbate reductase 2 (**K**), multiprotein-bridging factor (**L**).

**Figure 4 ijms-23-06012-f004:**
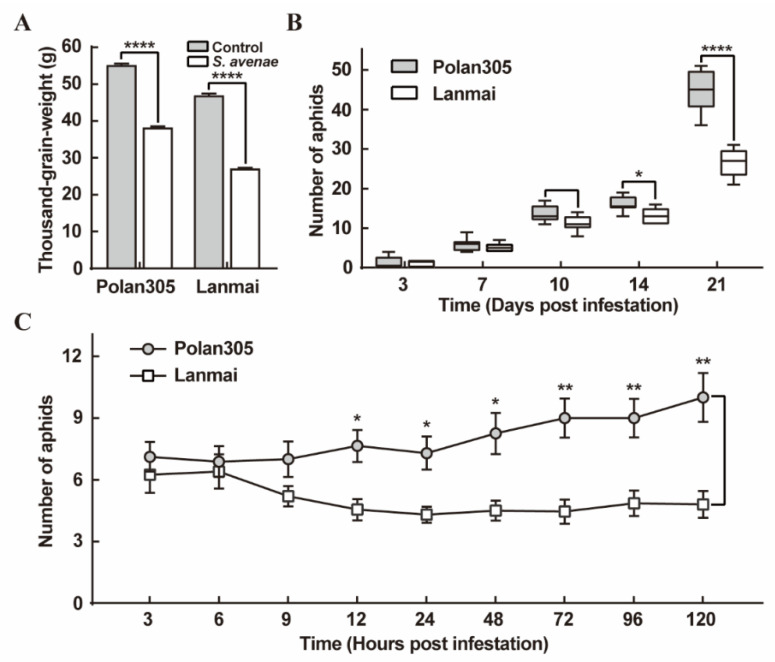
The performance of *S. avenae* on different wheat varieties. (**A**) The change of thousand-grain weight in resistant (Lanmai) and susceptible (Polan305) wheat varieties by *S. avenae* infestation. (**B**) The fecundity of *S. avenae* infesting on resistant (Lanmai) and susceptible (Polan305) seedlings. (**C**) *S. avenae* infestation preference between Lanmai and Polan305. Values are mean ± SE (*n* = 9). *, **, **** indicate significant differences at *p* < 0.05, *p* < 0.01, *p* < 0.0001 (two-way ANOVA followed by Sidak’s multiple comparisons test), respectively.

## Data Availability

All data supporting the findings of this study are available in the paper and its Appendix A published online. Further information may be obtained from the corresponding author, Wanquan Ji.

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
