# Peer review of "Identification of Differentially Expressed Genes in Resistant Tetraploid Wheat (Triticum turgidum) under Sitobion avenae (F.) Infestation"

_ijms, 2022, doi:10.3390/ijms23116012_

Round 1
Reviewer 1 Report
The study was focused on identification of differentially expressed genes in resistant tetraploid wheat (Triticum turgidum) under Sitobion avenae (F.) infestation. According to the Authors, suppression subtractive hybridization (SSH), a total of 134 ESTs were identified and categorized based on their putative functions. Quantitative RT-PCR analysis of 30 selected genes confirmed their differential expression over time between the resistant wheat cultivar Lanmai and susceptible wheat cultivar Polan305 during S. avenae infection. Importantly, there were eleven genes related to the photosynthesis process, and only three genes were observed to have higher expression in Lanmai than in Polan305 after S. avenae feeding. Gene expression analysis also revealed that Lanmai possessed a distinct regulatory effect on salicylic acid and jasmonic acid pathways after S. avenae infestation.
In my opinion the paper is quite interesting, however, I recommend the following improvements in order to increase its scientific soundness:
- Regarding Sitobion avenae, the word “infection” used in the manuscript should be replaced with “infestation”,
- Figure 3 is completely unreadable, and contains too large number of research data,
- Student's t-test is inappropriate for testing the set of data in the manuscript. A factorial ANOVA with a follow-up post-hoc test (Tukey's or Dunnet's) should be used,
- Some newer citations should be included, and the older ones may be discarded,
- Moderate English changes by the native speaker specialist are required.
Reviewer 2 Report
Dear Authors,
I have read your manuscript with interest. I believe that your research makes a necessary and important contribution to food security.
I advise you to improve the visual components of your manuscript. Figure 2 could be more descriptive. In the current version, you need to add explanations about color designations to the caption.
Figure 3 in its current form can be placed in the supplement, while a more descriptive infographic about this result should be added to the main part of the article.
Comments to the text are noted in the pdf file.
